# The relationship between visitors' motivation and landscape preference for the pilgrimage route on the Mount Miaofeng, China

Meijing Xu[1]ⓔ, Jianjiao Liu[2]ⓔ, Ru Wang[1]ⓔ, Shan Lu[1], Feng Xu[1]*

1 College of Horticulture, China Agricultural University, Beijing, China, 2 Faculty of Architecture, Building and Planning, University of Melbourne, Melbourne, Victoria, Australia

ⓔ These authors contributed equally to this work.
* ccxfcn@sina.com

**Data Availability Statement:** All relevant data are within the manuscript and its Supporting Information files.

**Funding:** This work was supported by the China Agricultural University [grant number 2017bj019].

## Abstract

Many countries have recognized the significance of religious tourist destinations and actively included them into tourism systems. This study took the pilgrimage route on the Mount Miaofeng, a famous sacred place in northern China, as the research site to comprehensively understand visitors' on-site experience when walking along it. The visitor-employed photography technique, supplemented by a questionnaire survey, was used to explore visitors' preferences for landscape elements and spatial sequences. The landscape elements were identified in eight categories (including vegetation, religious symbol, mountain, route, human, village, temple fair, and facility), and the spatial sequences of photos along the route were divided into containing views at close, middle, and remote distances. Visitors were classified into three types: those motivated by religious purposes, recreational purposes, and multiple purposes. It was concluded that visitors normally preferred the vegetative landscape and religious symbol categories, and they preferred to take photos with views at close distance (with cultural and aesthetic significance) in human settlements. Visitors motivated by different purposes showed different patterns of preferences for landscapes which are in consistent with their travel motivations. This study contributed to an in-depth interpretation of the people-place relationships and the enrichment of tourism motivation theory, and it could provide valuable references for the conservation, management, and planning of religious tourism destinations.

## Introduction

Religious heritages have been significant compositions of global tourism [1–5], and religious tourism is one of the core parts of the cultural tourism system. Derived from ancient times, religious tourism was defined as a form of tourism whereby people of faith travel individually or collectively for religious or spirituality-related reasons [6]. However, with the gradual integration of religious culture into tourism elements, the connotation and extension of religious tourism has been developing to also include sightseeing and leisure for non-religious purposes.

The funders had no role in study design, data collection and analysis, decision to publish, or preparation of the manuscript.

**Competing interests:** The authors have declared that no competing interests exist.

This evolution is reflected in the annual visitation of approximately 330 million people to major religious tourist destinations worldwide, as reported by the United Nations World Tourism Organization (UNWTO) [7, 8]. More and more individuals are going to tourist destinations for recreational purposes rather than just religious ones, thus there is an increasing trend towards secularization of religious tourist destinations.

The benefits of this secularization are significant, including the development and enhancement of religious and cultural knowledge as well as mutual understanding [9]. Therefore, the tourist destinations are not necessarily the point, especially for route-based pilgrimages which involve certain fixed routes, since the cultural and natural landscape along the routes and the travel experience itself are of equal importance [5, 10].

Studies have found that the religious tourism experience is the product of the correlation between secular tourism experience and sacred religious experience [11]. Research on religious travel experience emerged and developed in the 1960s, and many early scholars conducted research on the theory, significance, motivation, type, mechanism and quality of the tourist experience [12–15]. Visitors often appreciate the aesthetic value of the landscape while experiencing the spiritual journey, that is, the colors, shapes, textures, and other physical properties of the landscape [5]. History, art, and the daily lives of local people are also considered to be integral parts of the visitors' experience and are closely related to the authentic experience of seeking a heritage site [16].

Therefore, it is necessary to have an in-depth understanding of visitors' preferences, which are essential for tourism policy, management, and planning [17]. This study investigated visitors' travel motivations, their on-site experience, and landscape preferences, to further interpret people-place relationships, provide references for the conservation, management, planning of religious destinations, and enrich the tourism motivation theory. The pilgrimage route on the Mount Miaofeng in Beijing, which has a particularly long history and significant importance in the cultural and religious development of China, was selected as the research site. The main objectives of this study were to identify:

1. visitors' motivations for the pilgrimage route on Mount Miaofeng;

2. visitors' landscape preferences (landscape elements and spatial sequences) along the pilgrimage route;

3. the relationship between visitors' landscape preferences and their travel motivations.

## Literature review

### Religious tourism

The exploration of religious tourism mainly focused on the relationship between religion and tourism [18–21], the concept, resources, and influence of religious tourism [22–24], the development and optimization of scenic spots [25, 26], and the individual perception of visitors (such as the motivation, experience, satisfaction, etc.) [27–32]. Researchers such as Collins-Kreiner [18], MacCannell [19], Patel [20], and Ungureanu [21] have dedicated themselves to elucidating the intricate relationships between religion and tourism. Initially, there was a belief that religion and tourism were diametrically opposed concepts. However, this perspective was rejected by MacCannell in 1973 [19], leading to the current viewpoint of religion and tourism as a continuum developing in two directions. In this context, a visitor embodies both the characteristics of a tourist and a pilgrim.

Parallel to this, considerable attention has been paid to understanding the conceptual underpinnings, resource management, and the broader impacts of religious tourism. Iliev

[22], Kim et al. [23], and Aulet and Vidal [24] have provided insights into the operational and strategic aspects of managing religious tourism attractions. Their research highlights the importance of sustainable practices that enhance the visitors' experience without compromising the spiritual and cultural integrity of the sacred sites. Furthermore, studies by Mawarni and Puspitasari [25] and Sutianto et al. [26] have explored the optimization of scenic spots, suggesting that effective management of these areas leads to improved visitor satisfaction and can significantly influence the economy of the regions hosting these religious sites.

Moreover, researchers have focused on the individual perception of visitors, conducting exploratory studies on motivations, experience, and overall satisfaction with religious tourism. It is worth noting that some studies have shown that individual motives and experience play pivotal roles in shaping the demand for religious tourism [27–32]. They collectively agreed that the personal quest for spiritual fulfillment, cultural exploration, and historical appreciation significantly contribute to visitors' preferences and satisfaction, thus necessitating proper tourism management policies and landscape planning strategies to cater to these diverse visitor needs.

## Visitors' travel motivations in religious tourism

Research on visitors' travel motivations in religious tourism aligns with tourism motivation theory, which suggests that individuals travel to satisfy their spiritual or psychological needs [27, 33]. Tourism motivation can directly reflect visitors' needs, choice, and perceptions of certain destinations [34, 35]. For example, in tourism for medical purposes, the quality of medical service and healthcare products is positively related to visitors' satisfaction [36], while in tourism for festival purposes, seeking collective memories, experiencing the good life, and exploring cultural elements have a significant positive effect on perceived authenticity and satisfaction [37]. Research on visitors' motivations helps people better understand visitors' expectations and behaviors, which could be indicative of the conservation, management, and planning of these destinations.

Previous studies in the field of religious tourism have uncovered a variety of motivational factors driving visitors to sacred sites [28, 38–41]. These motivations can generally be categorized into three primary types: religious motivations, such as expressing strong religious belief or pray for luck; cultural motivations, such as visiting historical and cultural sites; and recreational motivations, such as getting away from everyday life and relaxing [42]. This diversity in motives highlights the multifaceted nature of religious tourism and underscores the need for a nuanced understanding of visitor behavior across different religious contexts.

## Landscape preference

Landscape preference also occupies a vital position in the whole process of tourism [43–45]. It refers to an observer's affinity for a specific landscape, which is discerned through comparative evaluation and represents the observer's definitive attitude towards that landscape. Consequently, landscape preference embodies the complex interactions between visitors and their surroundings. Urry [46] introduced the concept of the "tourist gaze", positing that visual perception (i.e., gaze) is central to the tourist experience. Tourists engage with landscapes through subjective lenses, shaping their experiences and preferences at each site. This gaze not only influences what visitors choose to observe but also how they document and interpret their experiences, particularly through photography.

Previous studies mainly focused on the visual (aesthetic) perceptions [47–49] and often utilized off-site photo-evaluation methods to reflect the experience in the real world. With the development of modern photographic techniques, visitor-employed photography (VEP) has been developed to capture the on-site dynamic interactions between people and the

environment [50–53]. First introduced by Cherem in 1972, the VEP technique allows visitors to take photos for specific environmental subjects without too much restriction from researchers' intentions [54]. Moreover, on-the-spot photos can better express visitors' real-time thoughts and reactions [43, 55]. Originally, the VEP process involved providing participants with inexpensive disposable cameras and asking them to take a set number of photographs based on personal choice or specific themes. Today, this method often utilizes visitors' own cameras or mobile devices, making it particularly effective for revealing insights into Chinese visitors' landscape perceptions. Photo-taking is an integral social practice among modern Chinese tourists and is generally not perceived as intrusive or difficult, requiring minimal prompting [56].

VEP provides a participatory approach to data collection, driven by the research subjects themselves, and has been used in the context of tourism measuring visitors' on-site environmental preferences, especially in terms of linear landscapes, such as corridors, trails, and rivers, with the effectiveness being validated by many studies [44, 52, 57]. Extensive studies utilizing user-generated content (UGC), such as travel photos, allows researchers to examine tourists' in situ behaviors. For instance, Ye et al. used the VEP method to investigate the on-site perceptions of Chinese tourists in Uluru-Kata Tjuta National Park, showcasing its potential for capturing authentic visitor experiences [56]. Similarly, Nielsen's study utilized VEP to assess visitors' perceptions of an urban forest by having volunteers capture images of their most and least favored locations along a trail in a near-urban recreational area in Gothenburg, Sweden [58]. These studies, alongside research on urban streetscapes [57], demonstrate the effectiveness of assessing visual attraction through UGC, further supporting the value of the VEP method in examining tourists' behaviors.

### Religious tourism in China

Research on religious tourism in China just started relatively late in recent decades, though the close connection between religion and tourism has been deeply rooted in the traditional Chinese culture [59, 60]. In March 2018, China's State Council adopted an institutional reform plan that merged the National Tourism Administration and the Ministry of Culture into the Ministry of Culture and Tourism. This important decision integrates culture and tourism, reflecting the Chinese government's emphasis on the cultural tourism market and pointing out the direction of future tourism development.

Much of the current research on religious tourism in China has focused on the cultural and historical values of specific sites or protection strategies [61, 62]. Some studies have investigated religious destinations such as the pilgrimage route on Mount Miaofeng, albeit from a folkloristic perspective, leaving visitors' travel motivations, on-site experience and preferences insufficiently explored [63–65]. Therefore, this research focused on the religious tourism in China and identify visitors' motivations, landscapes preferences (landscape elements and spatial sequences), and the relationship between visitors' landscapes preferences and their travel motivations along the pilgrimage route on Mount Miaofeng in China.

## Materials and methods

### Study area and population

The study was conducted in the Mount Miaofeng, a historical and cultural area in Beijing, China. The temples dedicated to the deity Bixia Yuanjun and Jade Emperor at the summit of the Moun Miaofeng have been recognized as famous sacred places that attract visitors from neighboring provinces and even neighboring countries such as Japan and Southeast Asian countries to undertake religious journeys. There were six pilgrimage routes to the summit of the Mount Miaofeng (Fig 1). The route from the Jiangou Village was the only access to the

most important sacred places–the Goddess Temple (also known as Huiji Temple, mainly for the deity Bixia Yuanjun) and the Peak of Jade Emperor (a temple complex mainly for the deity Jade Emperor) at the summit. Along this route, there are heritage sites such as Lingguan Temple and the Guansong Pavilion, as well as ancient pines of various shapes and sizes, with historical, cultural, and aesthetic values.

Therefore, the route from Jiangou Village to the Peak of Jade Emperor, a 1.65-kilometer section along the whole pilgrimage route, was chosen as the research site (Fig 1). It is the most well preserved and visited route section, and visitors like to travel on this section of pilgrimage route to show their devotion or to simply enjoy the scenery from the past to the present. People usually need an hour to walk the whole distance, which is an appropriate time length for this study. The targeted route went through diverse landscapes consisting of villages and forests, thus the route was divided accordingly based on the adjacent land-use types. It was finally divided into 17 visually distinguishable units of either forest landscape or human settlement (marked in light and dark blue, as shown in Fig 1). The on-site investigation was conducted on five fine weather days from May 12 to October 18, 2018, a time period considered to be the best travel season for the pilgrimage route.

This study followed the Declaration of Helsinki and was authorized by China Agricultural University's Human Body Research Ethics Committee (CAUHR-20230102). The participants were randomly selected from visitors who planned to walk through the study area. They were asked if they were willing to download the application of Six Foot, a travel-sharing and

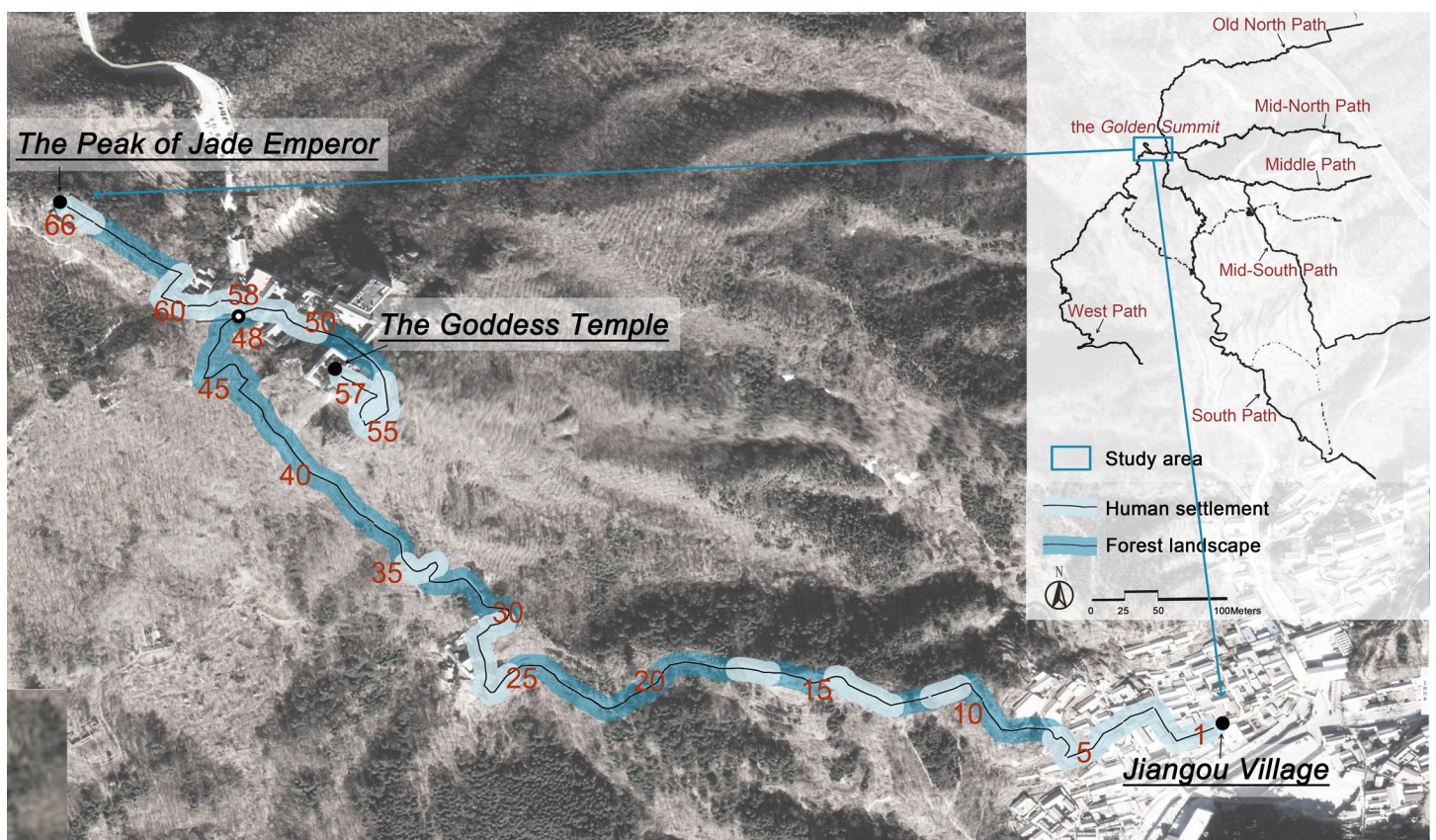

**Fig 1. The targeted pilgrimage route from Jiangou Village to the Peak of Jade Emperor of Mount Miaofeng.** The map image of Mount Miaofeng was obtained from OpenStreetMap (https://www.openstreetmap.org/edit#map=16/40.0661/116.0287) and is provided with an open access license.

footprint location software widely used by Chinese travel enthusiasts. This app could record visitors' travel track with a high degree of geographical accuracy using this application once they opened the location-based services (GPS) on their mobile phones. Only those who were planned to visit the targeted pilgrimage route from Jiangou Village to the Jade Emperor Peak, and those who were willing to download the Six Foot App were eventually included in this study. However, researchers may question the credibility of prior exposure to the research tool in the VEP method, as awareness of being part of a study might influence visitors' behavior, such as the number and types of photos taken. Nevertheless, as discussed in the literature review, relevant references on UGC allow researchers to examine tourists' behaviour and demonstrate the effectiveness of assessing visual preference and examining tourists' behaviors through VEP. All participants were informed of the research tasks and signed an informed consent form before the experiment.

## Data collection and analysis

Participants were asked to fill out a short questionnaire containing three sections: basic demographic information, travel-related characteristics of Mount Miaofeng, and a self-reported scale of travel motivations for visiting Mount Miaofeng (S1 File). The first section collected basic demographic information, including 6 items such as gender, city of region, age, religion, and interest in mountaineering or photography, adapted from Gou and Shibata [5]. Section two, adapted from Lee et al. [66], included 6 items such as the number of visits, frequency, companions, time spent, willingness for coming again, and duration of the trip. Section three comprised a 10-item self-reported scale of travel motivations for visiting Mount Miaofeng, adapted from Figler et al. [67]. Fig 2 provides a framework of this research.

**Visitors' travel motivations.** A self-reported scale was used to investigate visitors' travel motivations (see Table 1). The scale contains ten statements describing people's motivations for traveling to religious tourism destinations, such as "express strong religious beliefs", and "get away from everyday life and relax". Participants were asked to think about how true each statement was for them on a five-point Likert scale (1 = "strongly disagree" to 5 = "strongly agree"). The Cronbach's coefficient was 0.811, indicating high internal consistency and

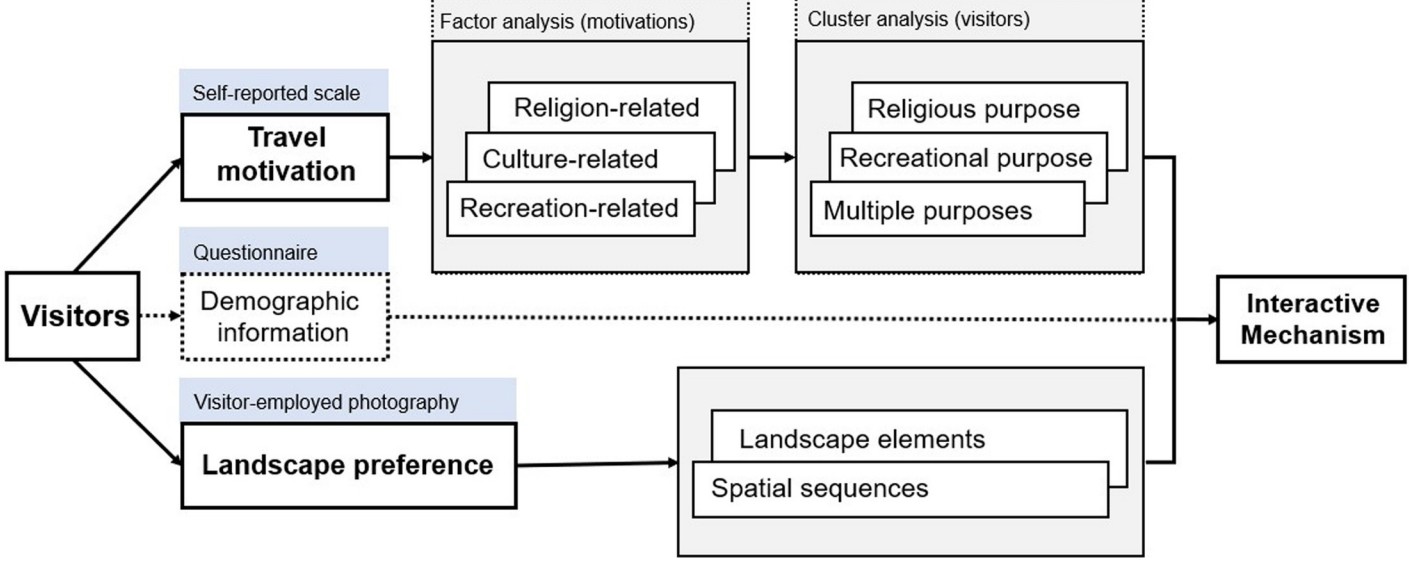

**Fig 2. A framework of the research.**

**Table 1. Categories and factor loading of visitors' motivations.**

| Categories | Motivation factor | Mean | SD | Factor loading |
|---|---|---|---|---|
| Religion-related motivation 3.10±0.93 | Express strong religious beliefs | 2.79 | 0.77 | 0.863 |
| | Worship religious deities | 2.85 | 0.86 | 0.887 |
| | Fulfill religious obligations | 2.76 | 0.78 | 0.650 |
| | Seek redemption and blessing to avoid disaster | 3.85 | 0.78 | 0.847 |
| | Pray for luck and repay wishes with appreciation | 3.24 | 0.96 | 0.841 |
| Culture-related motivation 3.35±0.78 | Broaden horizons and gain knowledge | 3.56 | 0.89 | 0.650 |
| | Satisfy curiosity about religious culture | 3.18 | 1.03 | 0.849 |
| | Appreciate religious architecture, literature, music and other arts | 3.32 | 1.04 | 0.803 |
| Recreation-related motivation **4.36±0.49** | Get away from everyday life and relax | **4.59** | 0.61 | 0.915 |
| | Experience local folklore | **4.12** | 0.84 | 0.538 |

reliability of the data. A factor analysis of the ten-factor motivation was conducted to further analyse the characteristics of visitors in terms of their travel motivations (see Table 1). This method was eligible in this case according to the KMO measure (0.698) and Bartlett's spherical test (sig = 0.000).

Then the cluster analysis was used to further categorize the participants into several groups according to their motivations. Ward's method was applied, and the Euclidean distance was selected to obtain a clustering tree diagram. The participants were divided into several major categories using the fast-clustering method (K-means clustering).

Thirty-four visitors (14 females, 20 males) were finally included in this research. Although the sample size is relatively small, it was deemed sufficient for several reasons. First, this study is exploratory (non-confirmatory), so a small sample size is appropriate. And the use of small sample sizes is typical in exploratory VEP research. For instance, Gou and Shibata included 31 participants [5], Li et al. included 30 participants [57], Fefer et al. had only 17 [47], and Markwell included just 15 participants [68]. This suggests that small sample sizes can already obtain enough information to yield effective research outcomes within the VEP field. Second, as Bonett suggested [69], for a single coefficient alpha test, assuming that the Cronbach's alpha coefficient equals zero in the null hypothesis can yield a minimum sample size of less than 30 to achieve a desired effect size of 0.7. While the general recommendation for sample sizes often suggests at least 100 respondents, Bonett also noted that for scales with a high expected alpha, a smaller sample size might be acceptable. In our study, the Cronbach's alpha coefficient is 0.811, and the minimum required sample size was calculated to be 34 (using the website: https://wnarifin.github.io/ssc/ssalpha.html, with the minimum acceptable Cronbach's alpha set at 0.6) [70]. Third, studies have shown that K-means clustering performs better with small sample sizes than with large ones. For example, Kumar and Dhamija applied the K-means clustering algorithm to real-life data with known cluster solutions and found that misclassification percentages increased with sample size, from small samples (n = 20 and n = 50) to larger samples (n = 100, 300, 500, 1000) [71]. Additionally, Henry et al. identified that K-means clustering can produce valid solutions with sample sizes as small as N = 20 [72], while another study demonstrated that the standard error of the average percentage of misclassification is lowest with the K-means method under small sample sizes (n = 30) [73]. Therefore, including 34 visitors in this research was deemed sufficient.

**Visitors' landscape preferences.** The method of visitor-employed photography (VEP) was adopted to capture visitors' on-site experience and their preferences for landscape along the route. Participants were asked to take photos while traveling along the targeted route using their mobile phones. At the same time, they needed to keep the Six Foot App in the mode of

"Map Travel" and open the location-based services. During this period, they could complete the trip at their own pace and take photos of whatever they wanted, without any right or wrong. In order to avoid interference, participants worked independently throughout the process without the accompany of investigators. The investigators waited for the participants at the end of the route, the Peak of Jade Emperor, assisted the participants in uploading Six Foot track and photo data, and then asked participants about reasons for photos they took (to get their on-site perceptions).

Photos were analysed through two perspectives–landscape elements and spatial sequences, to obtain visitors' preferences for the static content-based and dynamic context-related attributes of the route. They were also categorized according to visitors with different travel motivations to reveal the interactive mechanism.

*Landscape elements*. Qualitative coding was used to categorize photos from participants based on the types of landscape elements included in each photo (such as vegetation, road, religious symbol, etc.), combined with participants' reasons for taking photos, to identify participants' on-site perceptions for landscape elements. This coding process did not include a pre-set number of categories. Rather, categories were created through continually reviewing the photos and reflecting on the experiences of the participants. Photos could be assigned to different categories simultaneously according to the landscape elements they contained.

*Spatial sequences*. The photos were classified according to the distance of view, given the variability of landscape perception depends on the viewing perspective and the distance between landscape elements and the observer [74]. People can identify others' facial features at a distance of 30m, and identify their acquaintances through the characteristics of body features and movements at a distance of 50-100m; the morphology of trees can be recognized at a distance of 300-500m, while the canopy and main trunk is about 500-800m, and 3,000–5,000m for the forest landscape [75]. Therefore, the photographs were classified into three categories according to the range of view and the surrounding terrain. *Close distance*: photos containing landscape extending within 100m from the observer; *Middle distance*: 100-800m; *Remote distance*: more than 800m. As the exact location of each photo was known, the linear route was further divided into 66 sub-sections with 25-meter intervals using the ArcGIS software to facilitate the recording of the locations where the photos were taken. The route was divided into two directions in the sub-section 48 (Fig 1), southeast to the Goddess Temple (section 57) and northwest to the Peak of Jade Emperor (section 66). Then the number of photos at close, middle, and remote distances in each sub-section was calculated to reveal the perceived spatial sequences of the landscape.

## Results

The basic characteristics of the 34 visitors are shown in Table 2. The number of male participants was slightly higher than that of females. The majority of participants were young and middle-aged people from Beijing visiting the site for less than ten times.

### Visitors' travel motivations

Recreation-related motivation was the highest rated by visitors among three categories, with the highest mean score for the item "get away from everyday life and relax", followed by "experience local folklore", as shown in Table 1. "Seek redemption and blessing to avoid disaster", as well as "Pray for luck and repay wishes with appreciation" in the category of religion-related motivation were also highly rated, with other items in this category less significant, compared to items in the category of culture-related motivation.

**Table 2. Basic characteristics of visitors.**

| Variables | | Experimental group | Control group |
|---|---|---|---|
| Gender | Male | 10 | 10 |
| | Female | 7 | 7 |
| Origin of visitors | Beijing | 16 | 16 |
| | Other provinces | 1 | 1 |
| Age | 15–24 | 8 | 3 |
| | 25–44 | 7 | 11 |
| | 45–64 | 2 | 3 |
| Visits | Once | 10 | 5 |
| | 2–9 times | 5 | 10 |
| | More than 9 times | 2 | 2 |
| Time | Weekdays or weekends | 14 | 11 |
| | Folk temple fairs or Spring Festival | 5 | 3 |
| | Rose festival or red leaf festival | 1 | 4 |
| Characteristics | Mountaineering enthusiasts | 7 | 8 |
| | Photography enthusiasts | 4 | 7 |
| No. of photos per person | Max/ Min/ Mean | 38/ 1/ 10.4 | 25/ 1/ 5.8 |
| Duration along the route | Max/ Min/ Mean (min) | 66/ 30/ 49.8 | 70/ 29/ 42.6 |

**Note:** The folk temple fair refers to time periods when people make pilgrimages at the same time reach out to others or pray for blessings, dating back to ancient times, usually accompanied by traditional performances which have now become intangible cultural heritage. The rose festival and red leaf festival are newly developed time periods in the last decade or so, as a kind of publicity strategy to attract visitors.

Table 1 shows the results of factor analysis. Visitors' motivations were classified into three categories–religion-related, culture-related, and recreation-related motivations–out of ten items, and the mean value of each motivation was calculated. The factor loadings show the significance of the variables, with the higher the loading value, the stronger the correlation between the factors and the variables. Each motivation factor has a certain factor loading on multiple motivation types, indicating that visitors might have a combination of multiple motivations. The roles of pilgrim and visitor could be intertwined. Table 3 shows the results of cluster analysis which categorize the participants into three groups according to their motivations–visitors traveling mainly for religious purposes (13), multiple purposes (10), and recreational purposes (11). Visitors motivated by religious purposes normally wanted to

**Table 3. Classifications of visitors.**

| Motivation factor | Component | | | F | sig |
|---|---|---|---|---|---|
| | Religious purpose (13) | multiple purposes (10) | recreational purpose (11) | | |
| Express strong religious beliefs | 3 | 3 | 2 | 7.404 | 0.002 |
| Worship religious deities | 3 | 3 | 2 | 15.603 | 0.000 |
| Fulfill religious obligations | 3 | 3 | 2 | 16.749 | 0.000 |
| Seek redemption and blessing to avoid disaster | 4 | 4 | 4 | 1.931 | 0.162 |
| Pray for luck and repay wishes with appreciation | 4 | 3 | 3 | 10.820 | 0.000 |
| Broaden horizons and gain knowledge | 4 | 3 | 3 | 4.294 | 0.023 |
| Satisfy curiosity about religious culture | 4 | 4 | 2 | 13.864 | 0.000 |
| Appreciate religious architecture, literature, music and other arts | 4 | 4 | 2 | 11.358 | 0.000 |
| Get away from everyday life and relax | 5 | 4 | 5 | 1.085 | 0.350 |
| Experience local folklore | 4 | 3 | 4 | 13.663 | 0.000 |

express their religious beliefs and gain knowledge about religious culture in religious scenic spots, at the same time get away from the everyday life and relax their bodies and minds. Visitors with recreational purposes visited the Mount Miaofeng mainly to relax, experience the local folklore, and visit some of the special attractions of the scenic area. Visitors with multiple purposes aimed to make the most of the travel experience. While enjoying the visuals and relaxation, they also wanted to learn about the culture and experience the folklore and the everyday life conditions of the local area.

## Landscape elements

A total of 343 photos were taken by the visitors, among which 275 were valid photos (photographs that were unclear, repetitive, and not taken in the study area were excluded). Photographs were coded and categorized according to the landscape elements contained (Table 4). Since multiple landscape elements could be extracted from a single photograph, the sum of values in this column is the total number of landscape elements extracted from the photographs from each participant ($N_e = 422$).

The number of various landscape elements shown in the photos revealed that visitors payed the most attention to vegetation along the route, with 119 vegetation-related elements identified, representing 28.2% of the total elements (Table 4). The reasons for visitors to take photos of vegetation were multiple, such as "the red leaves in autumn are beautiful", "the climate and

**Table 4. Categories and descriptions of landscape elements in the photos.**

| Categories | Total | Descriptions | Sub-categories | Sub-total | Examples of reasons for taking photos |
|---|---|---|---|---|---|
| Vegetation | 119 (28.2%) | Forests with complex structures, ground cover plants, and ancient trees as landscape nodes | Forest canopy | 41 | "The red leaves in autumn are beautiful"; "great weather"; "the climate and vegetation are very different (from urban areas)"; "great natural environment"; "pine trees and pavilion are in harmony, like a Chinese painting"; "beautiful ancient pine trees" |
| | | | Ground cover | 40 | |
| | | | Ancient tree | 38 | |
| Religious symbol | 112 (24.4%) | Man-made tangible elements with religious significance, including stone monuments, sculptures and temples | Building | 49 | "Well-preserved ancient buildings and paintings"; "distinctive"; "the Peak of Jade Emperor is the highest"; "climbing to the Peak of Jade Emperor gives me a sense of accomplishment"; "the red walls and green tiles of the Goddess Temple are in harmony with pine trees" |
| | | | Structure | 37 | |
| | | | Sculpture | 17 | |
| | | | Other | 9 | |
| Mountain | 73 (17.3%) | A rolling, linear mountain range extending in a certain direction | Mountain range | **68** | "First visit to Mount Miaofeng"; "Mount Miaofeng is the closest mountain to Beijing"; "it's nice to look out over the horizon"; "very fairy-like"; "open views from higher ground" |
| | | | Natural Stone | 5 | |
| Route | 44 (10.4%) | The route itself and the surrounding environment | Stone step | 23 | "Love traces of history, like this mottled stone road"; "beautiful road"; "so tired of climbing" |
| | | | Stone road | 14 | |
| | | | Dirt road | 7 | |
| Human | 33 (7.8%) | Villagers, visitors and their activities | Villager | 6 | "Good weather for hiking and fitness" |
| | | | Visitor | 27 | |
| Village | 25 (5.9%) | Scenes of farmland, household and producing equipment around the village | Outside village | 17 | "There are many farmhouses in the villages down the mountain" |
| | | | Within village | 8 | |
| Temple fair | 13 (3.1%) | Temple activities including temple ceremonies, performances and visitor participation | Pilgrim association | 10 | "Very interested in the activities at Mount Miaofeng"; "I came to Mount Miaofeng during the temple fair to make a short film"; "the temple fair is a special feature of Mount Miaofeng and I wanted to record it"; "it feels like a very popular place" |
| | | | Recreation | 3 | |
| Facility | 3 (0.7%) | Road signs, restrooms, ticket offices, and rest facilities | / | 3 | "The number of seats is not enough" |

vegetation are very different (from urban areas)", and "beautiful ancient pine trees". It was closely followed by religious symbols (112, 24.4%), then mountains (73, 17.3%), and route itself (44, 10.4%), with facilities representing the least popular category. As for the sub-categories, mountain range obtained the most attention, with 68 elements identified, followed by buildings. Forest canopy, ground cover, ancient tree in the category pf vegetation, together with structure in religious symbols, roughly matched with each other in terms of the number of elements identified.

The preferences of visitors with different travel motivations for landscape elements were also looked into (Table 5). The average number of elements identified by visitors traveling for religious purposes was significantly higher of that for visitors traveling for recreational and multiple purposes. Religious symbols, vegetation and mountains were the most popular landscape elements, though with slight differences among three groups of visitors. Visitors with religious purposes and multiple purposes were more attracted by religious symbols and vegetation, which represented more than half of the total number in each group, followed by mountains. It was the vegetation and mountains that were more significant for recreational-purpose group, leaving other categories unrepresentative.

## Spatial sequences

Most photos were categorized as containing views at close distance (162), constituting 58.9% of the total number of photos, followed by 67 photos for middle distance, and 46 photos for remote distance, as shown in Fig 3.

Photos containing views at close distance (Fig 3A) were evenly distributed along the route, with several peaks, especially in the section 57 which reached a peak at the Peak of Jade Emperor. A higher number of photos (106, 65.4%) fell in the human settlement, most of which contained historical temples, pavilions, producing, and living facilities, in contrast to that in the forest. As for photos containing views at middle distance (Fig 3B), a slightly higher proportion of photos (58.2%) fell in the human settlement. The number of photos peaked at the section 66, where the Godden Summit was located. Another peak was also in the section 57, but the photos mainly contained spatial configurations of the Peak of Jade Emperor and surroundings, conveying a sense of space, compared to those for close distance which mainly focused on detailed landscape elements. The number of photos containing views at remote distance (Fig 3C) was the least, and the distribution of photos significantly clustered along the route, with 78.2% of photos falling in the human settlements. The number of photos reached a peak in the section 55, at the Godden Summit as well, and most of the photos were taken from a

**Table 5. Descriptive statistics of landscape elements in different categories of visitors.**

|  | Religious purpose | | Multiple purposes | | Recreational purpose | |
|---|---|---|---|---|---|---|
|  | **Mean** | **SD** | **Mean** | **SD** | **Mean** | **SD** |
| Vegetation | **3.923** | 3.343 | **3.000** | 3.134 | **3.455** | 8.408 |
| Religious symbol | **4.692** | 5.916 | **3.000** | 3.627 | 1.909 | 2.859 |
| Mountain | 1.923 | 2.362 | 1.700 | 2.497 | **2.818** | 2.786 |
| Route | 1.615 | 2.830 | 1.300 | 2.300 | 0.909 | 0.688 |
| People | 1.615 | 2.256 | 0.700 | 0.797 | 0.455 | 0.972 |
| Village | 1.308 | 1.514 | 0.500 | 0.897 | 0.273 | 0.488 |
| Fair | 0.769 | 1.051 | 0.000 | 0.316 | 0.273 | 0.849 |
| Facility | 0.077 | 0.399 | 0.200 | 0.560 | 0.000 | 0.000 |
| Total | **15.923** | 4.378 | 10.400 | 2.984 | 10.090 | 6.089 |

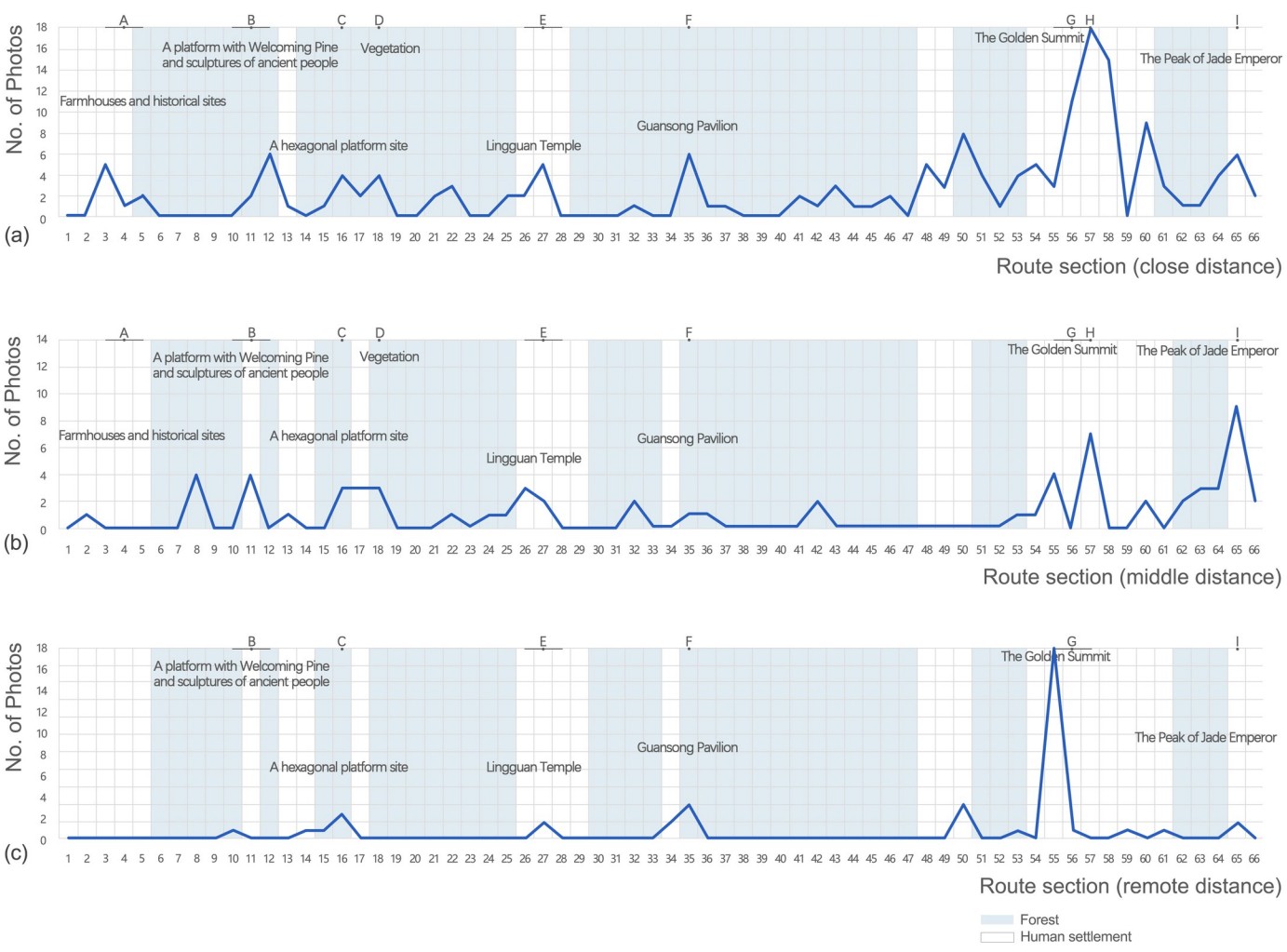

**Fig 3.** Characteristics of photos containing views (a) at close distance; (b) at middle distance; (c) at remote distance.

bird-view perspective presenting a whole structure of mountains, villages, and vegetation, since it was very high and provided a wide field of vision.

The preferences of visitors with different travel motivations for spatial sequences were analysed as well. The number of photos taken by visitors traveling for religious purposes was the highest (152) in all three distance categories, and the multiple purposes group (70) and recreational purposes group (63) had a similar number of photos. The number of photos containing views at close distance, compared to that at middle and remote distances, was the highest among all three groups of visitors, with a higher number of photos falling in the human settlements. Visitors traveling for religious purposes took significantly more photos with views at close distance, and the number of photos peaked in the section 57 at the Godden Summit (Fig 4A). The number of photos containing views at middle distance (Fig 4B) had similar distribution patterns among three groups of visitors. Photos were distributed evenly along the route, with several peaks, especially at the end of the route, such as section 57 and 65 (the Peak of Jade Emperor). There was also a higher proportion of photos falling in the human settlements. The number of photos containing views at remote distance (Fig 4C) was the least, and the distribution of photos significantly clustered in the human settlements along the route, especially in section 55, at the Godden Summit.

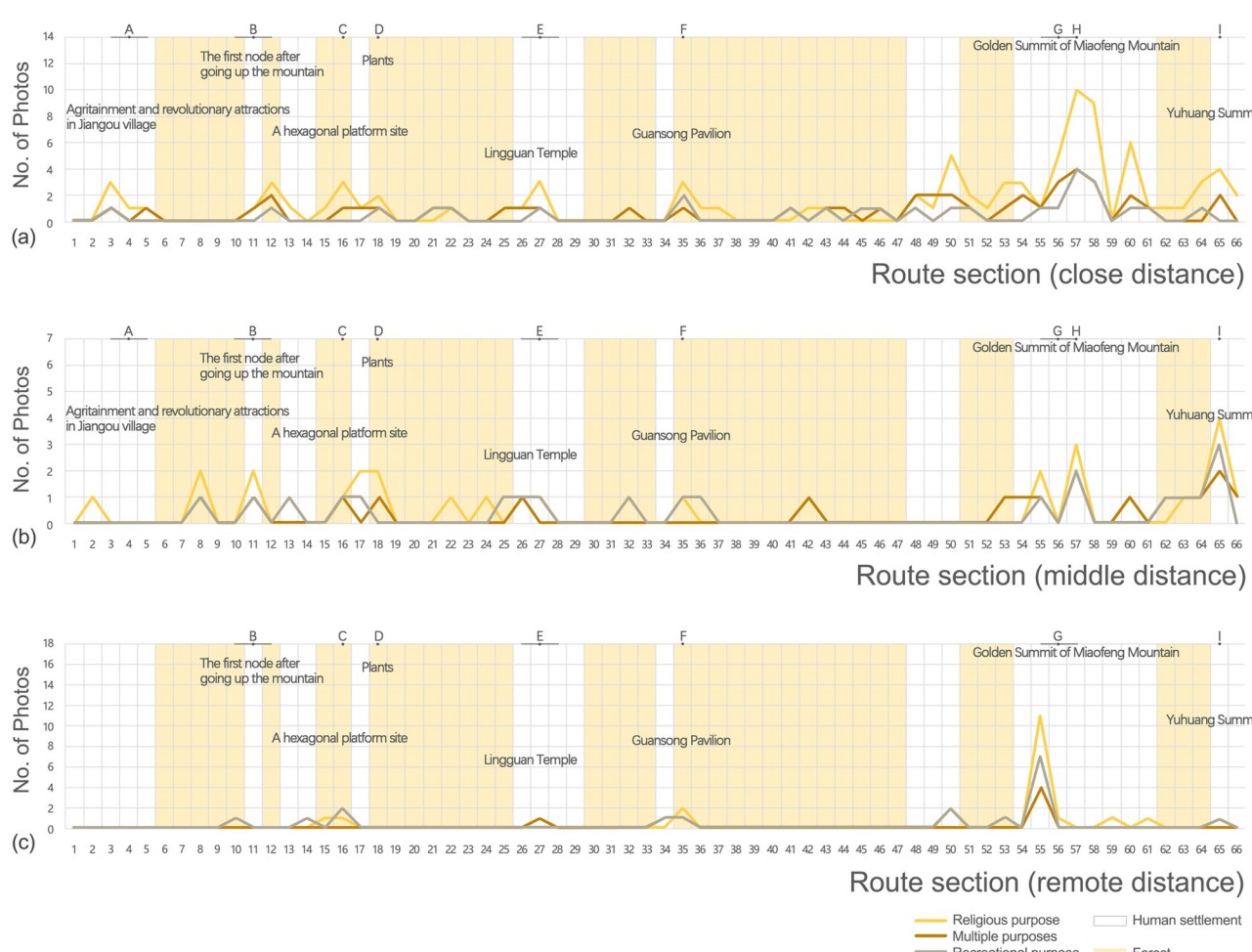

**Fig 4.** Characteristics of photos taken by three types of visitors containing views (a) at close distance; (b) at middle distance; (c) at remote distance.

## Discussion

### Religious purpose is no longer the primary motivation for religious tourism

Our results show that the recreation-related motivation was the highest rated by visitors among the three motivation categories (i.e. religion-related motivation, culture-related motivation, and recreation-related motivation), which suggested that religious purpose is no longer the primary motivation for religious tourism. Similar to existing studies [76–78], visitors in modern tourism destinations do not always have purely religious or spiritual motivations. For example, a 1998 survey of travellers in China found that 66% cited cultural rather than religious purposes [79]; Collins-Kreiner found that visitors' motivations for visiting the Holy Land were more secular, such as curiosity or recreation [18].

The change of the motivation of religious tourism is a symbol of the secularization of religious culture under the condition of modernity [80, 81]. The rapid development of modernization has led to the deterioration of urban environment, the emptiness of people's spirit, and the gradual decline of religious belief and architecture [81]. Therefore, people's purpose of religious travel is no longer to make a pilgrimage to the gods in their minds, but to enjoy the natural scenery of religious holy places, experience the mysterious religious culture, and

temporarily escape the negative consequences of modernization. The "divinity" of religious pilgrimage is decreasing continuously. The motivation and behavior mode of pilgrims are becoming increasingly secular, and the difference of tourism characteristics between pilgrims and tourists from other cultures is narrowing [80].

## Vegetative landscape and religious symbols, and photos with views at close distance were preferred

Visitors preferred vegetative landscape and religious symbols along the pilgrimage route. Vegetation makes up a large proportion of nature, and the "experience of nature" has been highly valued. Human beings had an "innately emotional affiliation" to nature–biophilia, originating in an understanding of human evolution [82], with a great body of posterior studies providing empirical support [83–87]. Religious symbols, referring to man-made tangible elements with religious significance, such as temples and monuments, highlighted the rarity and identity of the pilgrimage landscape. They could be recognized as cultural, historical, and religious landmarks [88, 89]. Visitors normally assessed the symbols as "distinctive", and they paid attention to the "well-preserved ancient buildings and paintings" and evaluated that "the red walls and green tiles of the Goddess Temple are in harmony with pine trees". Besides, other categories of landscape elements, such as mountain, route, people, village, and temple fair were also important for visitors' experience. This is probably because a combination of different elements could enrich the total environment and promote a range of interactions (e.g., purchasing local products or chatting with villagers or other visitors), contributing to a sense of place.

Photos containing views at close, middle and remote distances along the spatial sequences of the route could reflect visitors' preferences for detailed landscape attributes and spatial configurations. According to our results, the number of photos containing views at close distance was the highest, and most of them pictured historical temples, pavilions, ancient trees, producing, and living facilities. This indicated that visitors paid great attention to landscape elements and subtle details with cultural and aesthetic significance, that is, the content-based attributes of the environment. Photos in the middle and remote distance groups normally had no significant focal points, instead, it mainly pictured the spatial configurations of the landscape, evoking an awareness of here and there between visitors and their surroundings [58, 90]. High ground, such as the Godden Summit and the Peak of Jade Emperor had natural advantages of providing excellent line of sights and a wider range of vision, as well as positioning an unusual power to awaken a sense of the sacred. As visitors walking along the pilgrimage route, the spatial sequences with detailed landscape attributes and spatial configurations at close, middle, and remote distances alternately appeared, which combined to influence visitors' experience. In addition, most photos fell in human settlements instead of forests, supporting the importance of detailed landscape attributes with cultural and aesthetic values from a side view. The periodically appeared human settlements and forests had an important effect on enriching the spatial sequence as well, since monotonous or overly desolate walking environments were proved to be undesirable [45, 57, 58].

It is worth noting that visitors' sensory experience was also important. The reasons for taking photos were not only about the scenery, but their physical and emotional feelings. For example, they appreciated the "great weather", and they described their feelings as "climbing to the Peak of Jade Emperor gives me a sense of accomplishment". This is in consistent with the statement that non-visual experiences matter while walking along the route [91–93].

## Visitors' preference for landscape was in consistent with their travel motivations

The results suggested that visitors motivated by different purposes had different focuses on the landscape along the pilgrimage route. Visitors traveling for religious purposes showed significant preferences for religious symbol (e.g., temples), followed by vegetative landscape (e.g., ancient trees). They also paid attention to mountain, route, people, temple fair, etc., elements that were full of spiritual implications. They tended to take photos containing views at close distance with subtle details and content-based attributes which tell the stories and tales of the site, probably because of visitors' expectations to experience the spirit and culture of the specific site [5]. In contrast, visitors motivated by recreational purposes preferred landscapes with aesthetic value, such as vegetation and mountain, and the percentages of photos with views at close, middle, and remote distances were evenly distributed. The situation of visitors motivated by multiple purposes fell somewhere in between the above mentioned. They paid similar attention to vegetation and religious symbol, followed by mountain and route itself, and favoured close views as well.

At the same time, as visitors walking along the pilgrimage route, they gained more insights into the specific spaces [45]. This was not only because of the physical characteristics of the spaces, but also because walking along the pilgrimage route itself was an indispensable experience. Given that visitors motivated by different purposes have different preferences for landscapes along the pilgrimage route, it is important to incorporate appropriate landscape elements in proper sequences for management and planning [94–96]. Su et al. found that the more heterogeneous the landscape elements were throughout the latter part of the route, the more the visitors would be attracted to observe the route and surroundings [97].

## Conclusion

This study investigated visitors' travel motivations and preferences for landscape elements and spatial sequences along the pilgrimage route on the Mount Miaofeng. Eight categories of landscape elements were identified from photos taken by visitors and divided the spatial sequences of photos along the route into containing views at close, middle, and remote distances. The results illustrated that visitors normally preferred vegetative landscape and religious symbol. As for the spatial sequences, visitors preferred to take photos with views at close distance with cultural and aesthetic significance in human settlements. Visitors with different motivations (religious, recreational, or multiple purposes) showed different patterns of preferences for the landscape that are highly consistent with their travel motivations. Visitors motivated by religious purposes preferred religious symbol, followed by vegetation, and they took more photos with views at close distance. Visitors motivated by recreational purposes paid more attention to vegetation and mountain of highly aesthetic value, with evenly distributed numbers of photos picturing views at close, middle, and remote distances. The situation of visitors with multiple purposes fell somewhere in between the above mentioned two groups of visitors.

The results could serve as a reference for the conservation, management, and planning strategies for pilgrimage route or religious tourism, which should be based on a full understanding of visitors' preferences and motivations. For example, the conservation of the most important elements and route sections, such as the historical temples and ancient trees, could be prioritized. In terms of management and planning, visitors holding religious purposes could be provided with pilgrimage-related and route-specific information and guides, while visitors motivated by recreational purposes could be provided with various festivals, fairs, leisure activities, and so on. It is hoped that the sacred sites, such as the pilgrimage route in this study, would not be turned into either theme parks or museums with "authentic" heritages.

Therefore, it was recommended that managers should embrace the dynamic relationships between visitors and the specific environment, allowing diversified appreciations and interpretations landscape containing cultural, historical, and religious significance.

However, there are still some limitations in this research. First, as is typical for qualitative and place-related research, the results were confined to a limited group of respondents, making the study risky to generalize. Second, while investigating visitors on-site experience, the study focused primarily on the visual aspects of the perceptions, yet live landscapes appeal to all our senses. Smells and sounds experiences, for example, have been reported to be crucial to landscape experience [98, 99]. Third, visitors' demographic characteristics factors were not fully analysed in this study. Studies found that visitors' motivations varied in gender and age [28]. Male visitors were highly motivated by "curiosity and discovery", in contrast to females. Young people were less motivated by religion compared to the elderly. Furthermore, visitor photography techniques and questionnaires are commonly employed as research methods, they have their limitations. For example, visitor photography might be affected by individuals' photographic skills and interests, while questionnaires may be influenced by subjectivity and response preferences. Therefore, future research could expend the sample size and take demographic characteristics such as gender ratio, age, individual photographic skills and interests, travel patterns, and frequency into consideration. Notably, an important limitation of this study is that visitors were asked to download the Six Foot application. Being aware that they were part of a study might have influenced their behavior as visitors, including the number and types of photos they took. This aspect highlights the potential biases introduced by user awareness in user-generated content (UGC), which can affect how tourists' behaviors are examined. To address this limitation, a control group whose visitors were not aware of the study until they had completed their walk should be included in future research. Moreover, future studies can adopt more qualitative as well as quantitative methods to obtain richer contextual information and background knowledge to deeply understand the reasons and motivations behind their preferences for landscape elements and spatial sequences. Considering that visitors' travel motivations and landscape preferences may change over time, future long-term follow-up studies could be also conducted to better understand the travel motivations and landscape preferences of visitors.

## Supporting information

**S1 File. The questionnaire.**
(DOCX)

**S2 File. Data.**
(XLSX)

## Acknowledgments

The authors thank the participants in the experiment and the volunteers who helped conduct the experiment.

## Author Contributions

**Conceptualization:** Meijing Xu, Ru Wang.

**Data curation:** Meijing Xu, Jianjiao Liu, Ru Wang.

**Formal analysis:** Meijing Xu, Jianjiao Liu, Ru Wang.

**Funding acquisition:** Feng Xu.

**Investigation:** Jianjiao Liu, Ru Wang.

**Methodology:** Meijing Xu, Jianjiao Liu, Ru Wang.

**Resources:** Ru Wang.

**Software:** Meijing Xu, Ru Wang.

**Supervision:** Feng Xu.

**Validation:** Meijing Xu, Ru Wang, Shan Lu.

**Visualization:** Meijing Xu, Jianjiao Liu, Ru Wang.

**Writing – original draft:** Meijing Xu.

**Writing – review & editing:** Meijing Xu, Jianjiao Liu, Shan Lu, Feng Xu.

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
