## [Decision Letter · Decision Letter 0]

22 Mar 2024

PONE-D-24-00131The relationship between visitors' motivation and landscape preference for the pilgrimage route on the Mount Miaofeng, ChinaPLOS ONE

Dear Dr. Xu,

Thank you for submitting your manuscript to PLOS ONE. After careful consideration, we feel that it has merit but does not fully meet PLOS ONE’s publication criteria as it currently stands. Therefore, we invite you to submit a revised version of the manuscript that addresses the points raised during the review process.

**ACADEMIC EDITOR: This study seems interesting as it explore the relationship between visitors' motivation and landscape preference on the pilgrimage route. However, some demerits still need be improved. A major revision based on reviewers' comments is necessary for current manuscript.**

We look forward to receiving your revised manuscript.

Kind regards,

Qianda Zhuang, Ph.D.

Guest Editor

PLOS ONE

“This work was supported by the China Agricultural University [grant number 2017bj019].”

3. We note that your Data Availability Statement is currently as follows: [The original data presented in the study are included in the article/Supplementary material, further inquiries can be directed to the corresponding author.]

5. We note that Figure 3 in your submission contain copyrighted images. All PLOS content is published under the Creative Commons Attribution License (CC BY 4.0), which means that the manuscript, images, and Supporting Information files will be freely available online, and any third party is permitted to access, download, copy, distribute, and use these materials in any way, even commercially, with proper attribution. For more information, see our copyright guidelines: http://journals.plos.org/plosone/s/licenses-and-copyright.

a. You may seek permission from the original copyright holder of Figure 3 to publish the content specifically under the CC BY 4.0 license.

Reviewers' comments:

Reviewer's Responses to Questions

**Comments to the Author**

1. Is the manuscript technically sound, and do the data support the conclusions?

Reviewer #1: Partly

Reviewer #2: Yes

2. Has the statistical analysis been performed appropriately and rigorously? 

Reviewer #1: Yes

Reviewer #2: Yes

3. Have the authors made all data underlying the findings in their manuscript fully available?

Reviewer #1: Yes

Reviewer #2: Yes

4. Is the manuscript presented in an intelligible fashion and written in standard English?

Reviewer #1: Yes

Reviewer #2: Yes

5. Review Comments to the Author

Reviewer #1: To improve the work, so that it would be within the scientific framework, I propose several changes and suggestions:

-the paper must be written in the passive voice, for example, it should not be we used but it was used

-the paper should have an introduction and a literature review

-whose research was taken as a model for creating the questionnaire?

-proofreading is required

-there are a lot of technical errors

-in chapter 4.2 - use recent sources

-limitation of the paper need to transfer to chapter 5 (Conclusion) as well as recommendations for future research

-Figure 3 it is not clear (images are not visible and letters cannot be read)

Reviewer #2: This paper takes the pilgrimage route of Miaofeng Mountain, a famous sacred site in northern China, as the research site, to comprehensively understand the on-site experience of tourists walking on the pilgrimage route. Tourist photography techniques were used, supplemented by questionnaire surveys, to explore tourists' preferences for landscape elements and spatial sequences. This paper is helpful to deeply interpret the relationship between man and environment, enrich the theory of tourism motivation, and provide a reference for the protection, management and planning of religious tourism destinations.

1. The importance of research topic: Religious tourism, as a special form of tourism, is of great significance for understanding and explaining people's beliefs, cultures, and spiritual needs. This paper selects Miaofeng Mountain, a famous religious tourism destination in northern China, for research, which is not only helpful to understand the characteristics of religious tourism, but also provides a scientific basis for the protection, management and planning of religious tourism destinations.

2. Innovativeness of the research methodology: The paper adopts a combination of tourist photography technology and questionnaire survey, which can not only capture the real experience of tourists on site, but also gain insight into the preferences and motivations of tourists through questionnaires. In addition, the level and depth of research are further enriched by classifying the spatial sequences of photographs.

3. Theoretical contributions: This paper proposes a classification method for eight types of landscape elements and three types of spatial sequences, which provides a new perspective for the landscape research of religious tourism destinations. At the same time, this paper also conducts in-depth research on the landscape preference patterns of tourists with different purposes, which provides new empirical support for the theory of tourism motivation.

4. Practical guiding significance: The conclusions of this paper are of great guiding significance for the protection, management and planning of religious tourism destinations. For example, understanding tourists' preferences for vegetation landscapes and religious symbols can provide direction for the landscape design and maintenance of a destination; The study of the landscape preference patterns of tourists with different purposes can develop more precise marketing strategies for destinations.

5. Limitations and future prospects of the study:

Sample size and sample selection: There were 34 visitors, the sample size was sufficient, including 14 women and 20 men, and the ratio of men to women was somewhat unbalanced. Whether the tourists for different purposes of tourism are related to the age of the visitor. If the sample size is small or there is a lack of diversity, the generalizability and reliability of the findings may be compromised.

While visitor photography techniques and questionnaires are commonly used research methods, they also have limitations. For example, visitor photography may be influenced by individual photographic skills and interests, while questionnaires may be influenced by subjectivity and response preferences. If the sample size is sufficient, the error can be effectively reduced.

The abstract mentions tourists' preferences for landscape elements and spatial sequences, but does not further explore the reasons and motivations behind these preferences. Future research could provide an in-depth analysis of the psychological, social, and cultural factors behind tourist preferences to provide a more comprehensive explanation. Moreover, visitor behavior and preferences in religious tourism destinations may change over time, so studying long-term trends is essential to understand and predict the development of destinations.

6. PLOS authors have the option to publish the peer review history of their article (what does this mean?). If published, this will include your full peer review and any attached files.

Reviewer #1: No

Reviewer #2: No

---

## [Author Response · Author response to Decision Letter 0]

12 May 2024

PONE-D-24-00131

The relationship between visitors' motivation and landscape preference for the pilgrimage route on the Mount Miaofeng, China

PLOS ONE

Dear Editors, 

Thank you very much for editing our manuscript. We appreciate the encouragement and thoughtful suggestions from you and the reviewers.

We have carefully considered your comments and those of the reviewers, and revised our manuscript accordingly. We provided both a track-change version and an unmarked version for the main manuscript, and prepared a detailed list of our replies with line numbers listed.

The main modifications are summarized as follows:

1. Reviewed the provided templates and ensured the manuscript meets PLOS ONE’s style requirements, including those for file naming.

2. Added the following statement to the financial disclosure: “The funders had no role in study design, data collection and analysis, decision to publish, or preparation of the manuscript.”

3. Uploaded all raw data as a Supporting Information file (S2 File).

4. Supplied a replacement figure for Figure 1 that complies with the CC BY 4.0 license and removed the copyrighted images contained in Figure 3.

5. Rewrote in the passive voice, proofread, corrected technical errors, used recent resources in chapter 4.2, and ensured Figure 3 is visible with readable letters.

6. Revised the introduction, added a literature review section, and added a statement of creating the questionnaire.

7. Transferred limitation to the conclusion section and added more statements about the limitations and future prospects of the study as recommendations for future improvements.

We hope our revised version is suitable for publication in PLOS ONE. Thanks again for your editorial work and we are looking forward to hearing from you soon. 

Sincerely,

Feng Xu on behalf of all authors

Response to Academic Editor:

General comment: This study seems interesting as it explored the relationship between visitors’ motivation and landscape preference on the pilgrimage route. However, some demerits still need be improved. A major revision based on reviewers’ comments is necessary for current manuscript.

Comment 1: Please ensure that your manuscript meets PLOS ONE’s style requirements, including those for file naming. The PLOS ONE style templates can be found at

Response: Thanks for your comment. We have carefully reviewed the provided templates and have made the necessary adjustments to our manuscript to ensure it fully complies with PLOS ONE’s formatting guidelines. 

Comment 2: Thank you for stating the following financial disclosure:

“This work was supported by the China Agricultural University [grant number 2017bj019].”

Please state what role the funders took in the study. If the funders had no role, please state: “The funders had no role in study design, data collection and analysis, decision to publish, or preparation of the manuscript.”

Response: Thanks for your comment. We would like to clarify that the funders did not play any role in the study design, data collection and analysis, decision to publish, or preparation of the manuscript. As requested, we have added the following statement to the cover letter as financial disclosure to reflect this: “The funders had no role in study design, data collection and analysis, decision to publish, or preparation of the manuscript.”

Comment 3: We note that your Data Availability Statement is currently as follows: [The original data presented in the study are included in the article/Supplementary material, further inquiries can be directed to the corresponding author.]

Response: Thanks for your comment. To facilitate easy access and ensure transparency, we have decided to upload all raw data as a Supporting Information file (S2 File) along with our manuscript.

Comment 4: We note that Figure 1 in your submission contain [map/satellite] images which may be copyrighted. All PLOS content is published under the Creative Commons Attribution License (CC BY 4.0), which means that the manuscript, images, and Supporting Information files will be freely available online, and any third party is permitted to access, download, copy, distribute, and use these materials in any way, even commercially, with proper attribution. For these reasons, we cannot publish previously copyrighted maps or satellite images created using proprietary data, such as Google software (Google Maps, Street View, and Earth). For more information, see our copyright guidelines: http://journals.plos.org/plosone/s/licenses-and-copyright.

Please upload the completed Content Permission Form or other proof of granted permissions as an “Other” file with your submission.

Response: Thanks for your comment. Unfortunately, we are unable to obtain permission from the original copyright holder to publish the Google Map contained in Figure 1 under the CC BY 4.0 license. Therefore, we provided a replacement map figure from OpenStreetMap (https://www.openstreetmap.org/edit#map=16/40.0661/116.0287), which offers map data and shapefiles with open access licenses for Mount Miaofeng. This replacement map is compliant with the CC BY 4.0 license, and it replaces the copyrighted Google Map in Figure 1.

Comment 5: We note that Figure 3 in your submission contain copyrighted images. All PLOS content is published under the Creative Commons Attribution License (CC BY 4.0), which means that the manuscript, images, and Supporting Information files will be freely available online, and any third party is permitted to access, download, copy, distribute, and use these materials in any way, even commercially, with proper attribution. For more information, see our copyright guidelines: http://journals.plos.org/plosone/s/licenses-and-copyright.

a. You may seek permission from the original copyright holder of Figure 3 to publish the content specifically under the CC BY 4.0 license.

Please upload the completed Content Permission Form or other proof of granted permissions as an “Other” file with your submission.

Response: Thanks for your comment. We have removed the copyrighted images contained in Figure 3.

Response to Reviewer #1:

General comment: To improve the work, so that it would be within the scientific framework, I propose several changes and suggestions:

Comment 1: The paper must be written in the passive voice, for example, it should not be we used but it was used.

Response: Thanks for your comment. We have revised the paper, ensuring that descriptions are presented in the passive voice as suggested.

Comment 2: The paper should have an introduction and a literature review.

Response: Thanks for your comment. We have revised the introduction and added a literature review section (including four subsections: religious tourism, visitors’ travel motivations in religious tourism, landscape preference, and religious tourism in China) to better organize and clarify these sections.

Comment 3: Whose research was taken as a model for creating the questionnaire?

Response: Thanks for your comment. We have added extra explanations and citations in the revised manuscript to clarify the models for creating the questionnaire (line 209-218):

“Participants were asked to fill out a short questionnaire containing three sections: basic demographic information, travel-related characteristics of Mount Miaofeng, and a self-reported scale of travel motivations for visiting Mount Miaofeng (S1 File). The first section collected basic demographic information, including 6 items such as gender, city of region, age, religion, and interest in mountaineering or photography, adapted from Gou and Shibata [5]. Section two, adapted from Lee et al. [66], included 6 items such as the number of visits, frequency, companions, time spent, willingness for coming again, and duration of the trip. Section three comprised a 10-item self-reported scale of travel motivations for visiting Mount Miaofeng, adapted from Figler et al. [67].”

5. Gou S, Shibata S. Using visitor-employed photography to study the visitor experience on a pilgrimage route – A case study of the Nakahechi Route on the Kumano Kodo pilgrimage network in Japan. J Outdoor Recreat Tour. 2017 Jun 1;18:22–33.

66. Lee S, Lee S, Lee G. Ecotourists’ motivation and revisit intention: A case study of restored ecological parks in South Korea. Asia Pac J Tour Res. 2014 Nov 2;19(11):1327–44.

67. Figler MH, Weinstein AR, Sollers JJ, Devan BD. Pleasure travel (tourist) motivation: A factor analytic approach. Bull Psychon Soc. 1992 Aug 1;30(2):113–6.

Comment 4: Proofreading is required.

Response: Thanks for your comment. We have proofread this paper and checking for grammatical errors, typos, and formatting consistency.

Comment 5: There are a lot of technical errors.

Response: Thanks for your comment. We have proofread this paper and corrected technical errors.

Comment 6: In chapter 4.2 - use recent sources.

Response: Thanks for your comment. We have revisited this section and updated our references to include more recent sources to ensure the currency and relevance of the information presented.

Comment 7: Limitation of the paper need to transfer to chapter 5 (Conclusion) as well as recommendations for future research.

Response: Thanks for your comment. We have transferred the limitation to the conclusion section as well as recommendations for future research.

Comment 8: Figure 3 it is not clear (images are not visible and letters cannot be read).

Response: Thanks for your comment. Thanks for your comment and valuable feedback regarding the clarity of Figure 3 in our manuscript. We have checked the original image by downloading Figure 3 via the option located in the top right corner of the submitted PDF and found it should be clear. It is possible that the image was compressed during the PDF conversion process, which could have affected its clarity in the version you reviewed. Additionally, we have resubmitted the figure with the manuscript to ensure it meets the quality standards.

Response to Reviewer #2:

General comment: This p

---

## [Decision Letter · Decision Letter 1]

21 Aug 2024

PONE-D-24-00131R1The relationship between visitors' motivation and landscape preference for the pilgrimage route on the Mount Miaofeng, ChinaPLOS ONE

Dear Dr. Xu,

Thank you for submitting your manuscript to PLOS ONE. After careful consideration, we feel that it has merit but does not fully meet PLOS ONE’s publication criteria as it currently stands. Therefore, we invite you to submit a revised version of the manuscript that addresses the points raised during the review process.

**This paper need more revisions according to attached reviewers' comments. Please make sure their concerns are addressed for improving the quality. Please provide specific corrections or rebuttal point by point** **as feedback in the revision version.** Please submit your revised manuscript by Oct 05 2024 11:59PM. If you will need more time than this to complete your revisions, please reply to this message or contact the journal office at plosone@plos.org. Please include the following items when submitting your revised manuscript:A rebuttal letter that responds to each point raised by the academic editor and reviewer(s). You should upload this letter as a separate file labeled 'Response to Reviewers'.A marked-up copy of your manuscript that highlights changes made to the original version. You should upload this as a separate file labeled 'Revised Manuscript with Track Changes'.An unmarked version of your revised paper without tracked changes. You should upload this as a separate file labeled 'Manuscript'.If applicable, we recommend that you deposit your laboratory protocols in protocols.io to enhance the reproducibility of your results. Protocols.io assigns your protocol its own identifier (DOI) so that it can be cited independently in the future. For instructions see: https://journals.plos.org/plosone/s/submission-guidelines#loc-laboratory-protocols. Additionally, PLOS ONE offers an option for publishing peer-reviewed Lab Protocol articles, which describe protocols hosted on protocols.io. Read more information on sharing protocols at https://plos.org/protocols?utm_medium=editorial-email&utm_source=authorletters&utm_campaign=protocols.

We look forward to receiving your revised manuscript.

Kind regards,

Qianda Zhuang, Ph.D.

Guest Editor

PLOS ONE

Journal Requirements:

Reviewers' comments:

Reviewer's Responses to Questions

**Comments to the Author**

1. If the authors have adequately addressed your comments raised in a previous round of review and you feel that this manuscript is now acceptable for publication, you may indicate that here to bypass the “Comments to the Author” section, enter your conflict of interest statement in the “Confidential to Editor” section, and submit your "Accept" recommendation.

Reviewer #3: All comments have been addressed

Reviewer #4: (No Response)

2. Is the manuscript technically sound, and do the data support the conclusions?

Reviewer #3: Yes

Reviewer #4: Yes

3. Has the statistical analysis been performed appropriately and rigorously? 

Reviewer #3: Yes

Reviewer #4: Yes

4. Have the authors made all data underlying the findings in their manuscript fully available?

Reviewer #3: Yes

Reviewer #4: Yes

5. Is the manuscript presented in an intelligible fashion and written in standard English?

Reviewer #3: Yes

Reviewer #4: No

6. Review Comments to the Author

**Reviewer #3:** (No Response)

**Reviewer #4:** The title is descriptive and relevant to the aim of the article, while the abstract effectively summarizes the article's objectives, the methods used, and the main findings.

It is suggested that the authors extend the discussion on the literature review regarding the VEP. It would be interesting to start this discussion with Urry’s concept of the 'tourist gaze.' Typically, every article on visitor-employed photography begins by referencing this concept in one way or another.

The main concern relates to the sample size. It is suggested that the authors extend their explanation for using such a small sample size. For example, the minimum sample size for calculating Cronbach's alpha is generally considered to be at least 100 respondents. As Bonett (Bonett, D. G. (2002). Sample size requirements for testing and estimating coefficient alpha. Journal of educational and behavioral statistics, 27(4), 335-340.) suggested, for scales with a high expected alpha, a smaller sample size might be acceptable, but it is crucial to clearly explain why such a small sample size was deemed sufficient. Another reference might be: https://wnarifin.github.io/ssc/ssalpha.html. A similar discussion should be conducted from the perspective of K-means clustering. Overall, the sample size in this part of the research is a weakness of the article, and this concern must be carefully addressed.

The authors should acknowledge as a limitation of their study that visitors were asked to download the Six Foot application. Being aware that they were part of a study might have influenced their behavior as visitors, including the number and types of photos they took. A discussion of this limitation, either in the context of Materials and methods section, either within the literature review where the VEP (Visitor-Employed Photography) topic is addressed, could include relevant references on how user-generated content (UGC), such as travel photos, allows researchers to examine tourists' behavior.

7. PLOS authors have the option to publish the peer review history of their article (what does this mean?). If published, this will include your full peer review and any attached files.

Reviewer #3: No

Reviewer #4: No

---

## [Author Response · Author response to Decision Letter 1]

8 Oct 2024

Editorial Office 

PLOS ONE

Editors

Beijing, October 5th, 2024

 Revision for manuscript (PONE-D-24-00131 R1)

“The relationship between visitors’ motivation and landscape preference for the pilgrimage route on the Mount Miaofeng, China”

Dear Editors:

Thanks a lot for editing our manuscript entitled “The relationship between visitors’ motivation and landscape preference for the pilgrimage route on the Mount Miaofeng, China” (PONE-D-24-00131 R1). 

We appreciate the encouragement and thoughtful suggestions from you and the reviewers. We have carefully considered your comments and those of the reviewers, and revised our manuscript accordingly. We provided both a track-change version and an unmarked version for the main manuscript, and prepared a detailed list of our replies with line numbers listed.

We hope our revised version is suitable for publication in PLOS ONE. Thanks again for your editorial work and we are looking forward to hearing from you soon. 

Sincerely,

Feng Xu on behalf of all authors

Response to Reviewer #4:

General comment: The title is descriptive and relevant to the aim of the article, while the abstract effectively summarizes the article's objectives, the methods used, and the main findings.

Comment 1: It is suggested that the authors extend the discussion on the literature review regarding the VEP. It would be interesting to start this discussion with Urry’s concept of the “tourist gaze.” Typically, every article on visitor-employed photography begins by referencing this concept in one way or another.

Response: Thanks for your comment. We have extended the discussion in the literature review regarding the VEP by introducing Urry’s concept of the “tourist gaze.” This addition can be found in the revised manuscript (line 138-143):

“Consequently, landscape preference embodies the complex interactions between visitors and their surroundings. Urry [46] introduced the concept of the “tourist gaze”, positing that visual perception (i.e., gaze) is central to the tourist experience. Tourists engage with landscapes through subjective lenses, shaping their experiences and preferences at each site. This gaze not only influences what visitors choose to observe but also how they document and interpret their experiences, particularly through photography.”

46. Urry J. The Tourist Gaze: Leisure and Travel in Contemporary Societies. Sage Publications; 1990. 202 p. 

Additionally, we have extended the discussion in the literature review on VEP by detailing its evolution, as stated in the revised manuscript (line 151-157):

“Originally, the VEP process involved providing participants with inexpensive disposable cameras and asking them to take a set number of photographs based on personal choice or specific themes. Today, this method often utilizes visitors’ own cameras or mobile devices, making it particularly effective for revealing insights into Chinese visitors’ landscape perceptions. Photo-taking is an integral social practice among modern Chinese tourists and is generally not perceived as intrusive or difficult, requiring minimal prompting [57].”

57. Li C, Li P, Huang X. Liked and Disliked Streetscape Imagery: Relation to Emotional Motivation and Block Distribution From Tourist Bus Visitors. SAGE Open. 2022 Jul 1;12(3):21582440221117129. 

Comment 2: The main concern relates to the sample size. It is suggested that the authors extend their explanation for using such a small sample size. For example, the minimum sample size for calculating Cronbach's alpha is generally considered to be at least 100 respondents. As Bonett (Bonett, D. G. (2002). Sample size requirements for testing and estimating coefficient alpha. Journal of educational and behavioral statistics, 27(4), 335-340.) suggested, for scales with a high expected alpha, a smaller sample size might be acceptable, but it is crucial to clearly explain why such a small sample size was deemed sufficient. Another reference might be: https://wnarifin.github.io/ssc/ssalpha.html. A similar discussion should be conducted from the perspective of K-means clustering. Overall, the sample size in this part of the research is a weakness of the article, and this concern must be carefully addressed.

Response: Thanks for your comment. We have provided a more detailed explanation for using a small sample size and included the following statement in the revised manuscript (lines 259-281):

“Although the sample size is relatively small, it was deemed sufficient for several reasons. First, this study is exploratory (non-confirmatory), so a small sample size is appropriate. And the use of small sample sizes is typical in exploratory VEP research. For instance, Gou and Shibata included 31 participants [5], Li et al. included 30 participants [56], Fefer et al. had only 17 [47], and Markwell included just 15 participants [68]. This suggests that small sample sizes can already obtain enough information to yield effective research outcomes within the VEP field. Second, as Bonett suggested [69], for a single coefficient alpha test, assuming that the Cronbach’s alpha coefficient equals zero in the null hypothesis can yield a minimum sample size of less than 30 to achieve a desired effect size of 0.7. While the general recommendation for sample sizes often suggests at least 100 respondents, Bonett also noted that for scales with a high expected alpha, a smaller sample size might be acceptable. In our study, the Cronbach’s alpha coefficient is 0.811, and the minimum required sample size was calculated to be 34 (using the website: https://wnarifin.github.io/ssc/ssalpha.html, with the minimum acceptable Cronbach’s alpha set at 0.6) [70]. Third, studies have shown that K-means clustering performs better with small sample sizes than with large ones. For example, Kumar and Dhamija applied the K-means clustering algorithm to real-life data with known cluster solutions and found that misclassification percentages increased with sample size, from small samples (n=20 and n=50) to larger samples (n=100, 300, 500, 1000) [71]. Additionally, Henry et al. identified that K-means clustering can produce valid solutions with sample sizes as small as N=20 [72], while another study demonstrated that the standard error of the average percentage of misclassification is lowest with the K-means method under small sample sizes (n=30) [73]. Therefore, including 34 visitors in this research was deemed sufficient.”

5. Gou S, Shibata S. Using visitor-employed photography to study the visitor experience on a pilgrimage route – A case study of the Nakahechi Route on the Kumano Kodo pilgrimage network in Japan. J Outdoor Recreat Tour. 2017 Jun 1;18:22–33. 

47. Fefer JP, Hallo JC, Dvorak RG, Brownlee MTJ, Collins RH, Baldwin ED. Pictures of polar bears: Using visitor employed photography to identify experience indicators in the Arctic National Wildlife Refuge. J Environ Manage. 2020 Sep 1;269:110779. 

56. Ye IQ, Hughes K, Walters G, Mkono M. Up close and personal: Using high engagement techniques to study Chinese visitors’ landscape perceptions. Tour Manag Perspect. 2020 Jan 1;33:100629. 

68. Markwell KW. Dimensions of photography in a nature-based tour. Ann Tour Res. 1997 Jan 1;24(1):131–55. 

69. Bonett DG. Sample Size Requirements for Testing and Estimating Coefficient Alpha. J Educ Behav Stat [Internet]. 2002 Dec 1 [cited 2024 Sep 25]; Available from: https://journals.sagepub.com/doi/abs/10.3102/10769986027004335

70. Arifin W. Sample size calculator (web) [Internet] [Internet]. 2024 [cited 2024 Oct 4]. Available from: http://wnarifin.github.io

71. Kumar UA, Dhamija Y. Comparative analysis of SOM neural network with K-means clustering algorithm. In: 2010 IEEE International Conference on Management of Innovation & Technology [Internet]. 2010 [cited 2024 Sep 26]. p. 55–9. Available from: https://ieeexplore.ieee.org/abstract/document/5492838

72. Henry D, Dymnicki AB, Mohatt N, Allen J, Kelly JG. Clustering Methods with Qualitative Data: a Mixed-Methods Approach for Prevention Research with Small Samples. Prev Sci. 2015 Oct 1;16(7):1007–16. 

73. Wahi S, Dash S, Rao A. An Empirical Investigation on Classical Clustering Methods. 2009 Aug 10; 

Comment 3: The authors should acknowledge as a limitation of their study that visitors were asked to download the Six Foot application. Being aware that they were part of a study might have influenced their behavior as visitors, including the number and types of photos they took. A discussion of this limitation, either in the context of Materials and methods section, either within the literature review where the VEP (Visitor-Employed Photography) topic is addressed, could include relevant references on how user-generated content (UGC), such as travel photos, allows researchers to examine tourists’ behavior.

Response: Thanks for your comment. We have addressed the limitation regarding visitors being asked to download the Six Foot application in both the literature review and the Materials and Methods sections, and we have cited relevant references on how user-generated content (UGC), such as travel photos, allows researchers to examine tourists’ behaviour. 

In the literature review (lines 161-170), we included: 

“Extensive studies utilizing user-generated content (UGC), such as travel photos, allows researchers to examine tourists’ in situ behaviors. For instance, Ye et al. used the VEP method to investigate the on-site perceptions of Chinese tourists in Uluru-Kata Tjuta National Park, showcasing its potential for capturing authentic visitor experiences [56]. Similarly, Nielsen’s study utilized VEP to assess visitors’ perceptions of an urban forest by having volunteers capture images of their most and least favored locations along a trail in a near-urban recreational area in Gothenburg, Sweden [58]. These studies, alongside research on urban streetscapes [57], demonstrate the effectiveness of assessing visual attraction through UGC, further supporting the value of the VEP method in examining tourists’ behaviors.”

56. Ye IQ, Hughes K, Walters G, Mkono M. Up close and personal: Using high engagement techniques to study Chinese visitors’ landscape perceptions. Tour Manag Perspect. 2020 Jan 1;33:100629. 

57. Li C, Li P, Huang X. Liked and Disliked Streetscape Imagery: Relation to Emotional Motivation and Block Distribution From Tourist Bus Visitors. SAGE Open. 2022 Jul 1;12(3):21582440221117129. 

58. Nielsen AB, Heyman E, Richnau G. Liked, disliked and unseen forest attributes: Relation to modes of viewing and cognitive constructs. J Environ Manage. 2012 Dec 30;113:456–66.

In the Materials and Methods section (lines 226-231), we stated: 

“However, researchers may question the credibility of prior exposure to the research tool in the VEP method, as awareness of being part of a study might influence visitors’ behavior, such as the number and types of photos taken. Nevertheless, as discussed in the literature review, relevant references on UGC allow researchers to examine tourists’ behaviour and demonstrate the effectiveness of assessing visual preference and examining tourists’ behaviors through VEP.” In order to minimize disturbance to visitors and make the photos taken reflect their real behavior, two measures have been taken. First, we only inform visitors that they could complete the trip at their own pace and take photos of whatever they wanted, without any right or wrong (lines 286-287). Second, participants worked independently throughout the process without the accompany of investigators (lines 287-289).

Additionally, we explicitly acknowledged this limitation in the conclusion (lines 545-550):

“Notably, an important limitation of this study is that visitors were asked to download the Six Foot application. Being aware that they were part of a study might have influenced their behavior as visitors, including the number and types of photos they took. This aspect highlights the potential biases introduced by user awareness in user-generated content (UGC), which can affect how tourists’ behaviors are examined. To address this limitation, a control group whose visitors were not aware of the study until they had completed their walk should be included in future research.”

---

## [Decision Letter · Decision Letter 2]

7 Nov 2024

The relationship between visitors' motivation and landscape preference for the pilgrimage route on the Mount Miaofeng, China

PONE-D-24-00131R2

Dear Dr. Xu,

We’re pleased to inform you that your manuscript has been judged scientifically suitable for publication and will be formally accepted for publication once it meets all outstanding technical requirements.

Kind regards,

Qianda Zhuang, Ph.D.

Guest Editor

PLOS ONE

Additional Editor Comments (optional):

Reviewers' comments:

Reviewer's Responses to Questions

**Comments to the Author**

1. If the authors have adequately addressed your comments raised in a previous round of review and you feel that this manuscript is now acceptable for publication, you may indicate that here to bypass the “Comments to the Author” section, enter your conflict of interest statement in the “Confidential to Editor” section, and submit your "Accept" recommendation.

Reviewer #4: (No Response)

Reviewer #5: (No Response)

2. Is the manuscript technically sound, and do the data support the conclusions?

Reviewer #4: (No Response)

Reviewer #5: Yes

3. Has the statistical analysis been performed appropriately and rigorously? 

Reviewer #4: (No Response)

Reviewer #5: Yes

4. Have the authors made all data underlying the findings in their manuscript fully available?

Reviewer #4: Yes

Reviewer #5: Yes

5. Is the manuscript presented in an intelligible fashion and written in standard English?

Reviewer #4: Yes

Reviewer #5: Yes

6. Review Comments to the Author

Reviewer #4: Dear Authors,

You managed to address all the suggestions and comments from the first phase of the reviewing process.

Reviewer #5: Upon reviewing this version of the paper, I believe it is largely well-prepared for publication, as it effectively addresses the core objectives and contributes valuable insights to the field. However, I noted a couple of minor areas that may benefit from further refinement:

1. Flow and Sentence Structure: Some sections, particularly the literature review, contain lengthy sentences that may impact readability. Breaking down complex sentences can enhance clarity and ensure that key points stand out for the reader.

2. The discussion on Visitor-Employed Photography (VEP) and its evolution is insightful, but it might benefit from additional context about its significance in similar studies, if possible, with more recent references.

Congratulations to the authors.

7. PLOS authors have the option to publish the peer review history of their article (what does this mean?). If published, this will include your full peer review and any attached files.

Reviewer #4: No

Reviewer #5: No

---

## [Editor Report · Acceptance letter]

19 Nov 2024

PONE-D-24-00131R2 

PLOS ONE

Dear Dr. Xu, 

I'm pleased to inform you that your manuscript has been deemed suitable for publication in PLOS ONE. Congratulations! Your manuscript is now being handed over to our production team.

Kind regards, 

on behalf of

Dr. Qianda Zhuang 

Guest Editor

PLOS ONE